# PPARγ Targets-Derived Diagnostic and Prognostic Index for Papillary Thyroid Cancer

**DOI:** 10.3390/cancers13205110

**Published:** 2021-10-12

**Authors:** Jaehyung Kim, Soo Young Kim, Shi-Xun Ma, Seok-Mo Kim, Su-Jin Shin, Yong Sang Lee, Hojin Chang, Hang-Seok Chang, Cheong Soo Park, Su Bin Lim

**Affiliations:** 1Department of Biochemistry and Molecular Biology, Ajou University School of Medicine, Suwon 16499, Korea; stride@ajou.ac.kr; 2Department of Surgery, Ajou University School of Medicine, Suwon 16499, Korea; kimsuy@aumc.ac.kr; 3Department of Neurology, Institute for Cell Engineering, Johns Hopkins University School of Medicine, Baltimore, MD 21205, USA; shixun625@jhmi.edu; 4Thyroid Cancer Center, Department of Surgery, Institute of Refractory Thyroid Cancer, Gangnam Severance Hospital, Yonsei University College of Medicine, Seoul 06273, Korea; medilys@yuhs.ac (Y.S.L.); docjang@yuhs.ac (H.C.); surghsc@yuhs.ac (H.-S.C.); 5Department of Pathology, Gangnam Severance Hospital, Yonsei University College of Medicine, Seoul 06273, Korea; charm@yuhs.ac; 6CHA Ilsan Medical Center, Department of Surgery, Goyang-si 10414, Korea; sohopeacock@naver.com

**Keywords:** machine learning, prognosis, diagnosis

## Abstract

**Simple Summary:**

Through targeted next-generation sequencing of thyroid cancer-related genes in monozygotic twins with papillary thyroid cancer (PTC), we identified common variants of the gene encoding peroxisome proliferator activated receptor gamma (*PPARG*). Notably, the expression levels of PPARγ target genes were frequently deregulated in PTC compared to benign tissues and were closely associated with disease-specific survival (DSS) outcomes in a TCGA-PTC cohort. Machine learning-powered personalized scoring index comprising 10 PPARγ targets, termed as PPARGi, achieved a near-perfect accuracy in distinguishing cancers from benign tissues, and further identified a small subpopulation of patients at high-risk across different profiling platforms.

**Abstract:**

In most cases, papillary thyroid cancer (PTC) is highly curable and associated with an excellent prognosis. Yet, there are several clinicopathological features that lead to a poor prognosis, underscoring the need for a better genomic strategy to refine prognostication and patient management. We hypothesized that PPARγ targets could be potential markers for better diagnosis and prognosis due to the variants found in *PPARG* in three pairs of monozygotic twins with PTC. Here, we developed a 10-gene personalized prognostic index, designated PPARGi, based on gene expression of 10 PPARγ targets. Through scRNA-seq data analysis of PTC tissues derived from patients, we found that PPARGi genes were predominantly expressed in macrophages and epithelial cells. Machine learning algorithms showed a near-perfect performance of PPARGi in deciding the presence of the disease and in selecting a small subset of patients with poor disease-specific survival in TCGA-THCA and newly developed merged microarray data (MMD) consisting exclusively of thyroid cancers and normal tissues.

## 1. Introduction

The worldwide incidence of thyroid cancer has been rising rapidly in the past three decades, with the largest contribution being in papillary thyroid cancer (PTC) in all countries analyzed in a global population-based assessment study [1]. While it remains to be investigated whether the over-diagnosis is attributed to increased screening programs, South Korea has the world’s highest rate of thyroid cancer, which increased 15-fold between 1993 and 2011 [2]. Nevertheless, a 10-year disease-specific survival (DSS) by the American Joint Committee on Cancer/Union for International Cancer Control (AJCC/UICC) defined stages I to IV ranging from 81% (stage IV) to 100% (stage II) [3]. Despite being considered as an indolent tumor with low occurrence of local invasion, recurrences and regional or distant metastases, PTC, similar to other types of cancer, show inter- and intra-tumoral heterogeneity with varying degree of genetic diversity, which could have a significant impact on prognosis and on the response to targeted therapy [4,5]. The existence of a small population of tumors having the more aggressive variants of PTC, with distinct clinical, pathological, and molecular features suggest the need for a robust predictive marker for patient stratification disease management [4,5,6].

In thyroid cancer, main driver genetic alterations include *BRAF* (V600E) and *RAS* (NRAS involving codons 12 and 61) mutations, *RET* gene fusions, and *PAX8-PPARG* gene fusions [7,8]. A PAX8-PPARγ fusion protein (PPFP) is produced when the promoter and most of the *PAX8* gene, which encodes an important transcription factor for normal thyroid gland development, fuse with the coding exons of PPARγ, a member of the steroid/thyroid nuclear receptor family [8,9]. PPFP is found in about one-third of follicular thyroid carcinomas (FTC) and can act as an oncoprotein, as evidenced by in vivo and in vitro studies [8,10,11].

Similar to other subtypes of PPAR, PPARγ forms heterodimers with retinoid X receptor alpha (RXRα) and binds to specific DNA sequences, termed as peroxisome proliferator response elements (PPREs), in a ligand-responsive manner to trans-activate target genes [12]. Previous studies have revealed cell-specific maps and 3D structure of the intact PPARγ-RXRα complex, including binding sites within gene promoters and intergenic or intronic regions [12,13,14]. Variants in the *PPARG* gene were found to reduce the receptor-binding affinity to the PPREs and modulate transcriptional activity of its target genes, such as acyl coenzyme A (acyl-CoA) oxidase, regulating insulin sensitivity and adipose tissue differentiation [15].

While PPARγ is expressed at extremely low levels in the normal thyroid [11], its expression is highly induced in a variety of immune cells, including monocytes and macrophages, in addition to the well-studied adipocytes, governing cellular phenotype and function such as lipid metabolism and secretome through transactivation of PPARγ target genes [11,16,17]. Several key molecular mechanisms regulated by PPARγ in macrophages include differentiation [18], M1-to-M2 polarization [19], lipid metabolism [20], suppression of the production of pro-inflammatory cytokines (e.g., TNFα, IL-1B, and IL-6), and the expression of inflammation-related genes (e.g., iNOS and MMP9) [8,21].

Here, we hypothesized that the PPARγ target genes could serve as a prognosticator of outcome in patients with PTC due to the common variants found in an intronic region located between exon 5 and exon 6 of the *PPARG* gene encoding the ligand-binding domain (LBD) of nuclear hormone—a key domain for transactivation of PPARγ targets [11,22]—in three pairs of monozygotic twins harboring PTC. A newly developed personalized scoring index computed based on the expression levels of PPARγ target genes applied to merged microarray data (MMD) consisting exclusively of thyroid cancers and normal tissues and TCGA-THCA (thyroid carcinoma) data revealed its robust diagnostic and prognostic ability in predicting tissue-based disease-specific survival (DSS). Leveraging single-cell RNA sequencing (scRNA-seq) data and machine learning algorithms, we further identified specific cell types contributing predominantly to the PPARGi and demonstrated clinical significance and applicability of PPARGi assay in a routine clinical setting, regardless of profiling platform.

## 2. Materials and Methods

### 2.1. Subjects

We performed a retrospective study targeting identical twins who were treated with surgery for papillary thyroid cancer at Gangnam Severance Hospital, Yonsei University College of Medicine, between May 2009 and March 2020. Three pairs of identical twins with papillary thyroid carcinoma, denotated as TP1, TP2, and TP3 were identified. All methods were carried out in accordance with relevant guidelines and regulations. This study protocol was approved by the Institutional Review Board of Yonsei University (IRB 3-2020-0281). The IRB of Yonsei University waived the requirement for patient-informed consent as the study is retrospective by design. The clinicopathologic characteristics of the patients and tumors including age, sex, operation date, surgical extent, tumor size, and central and lateral lymph node metastasis are summarized in Appendix A.

### 2.2. Targeted DNA Sequencing and Analysis

Genomic DNA was extracted from FFPE-fixed tumor tissues. Sequencing libraries were prepared by Macrogen (Seoul, Korea) using SureSelect Target Enrichment kit (Agilent Technologies). Two distinct target panels were designed to detect fusion genes and mutations in coding exons (Appendix A) using SureSelect Custom DNA Target Enrichment Probes. The libraries were subjected to Illumina platform in paired-end (2 × 150 bp) mode. Analytical platforms used by Macrogen include (1) FASTQC, fastp (quality check and trimming); (2) BWA, PICARD, SAMTOOLS, and BEDTOOLS (alignment); and (3) MuTect2 (GATK) and LUMPY (variant calling). The human assembly GRCh37/hg19 was used for reference genome. The variant call format (VCF v4.2) provided for each sample by Macrogen was used to identify variants (annotated with SnpEff v4.3), in which only the passing variants annotated as PASS were considered as true variants.

### 2.3. Mapping of Protein Mutation Data

MutationMapper tool [23] provided by cBioPortal (cbioportal.org/mutation_mapper; accessed on 6 April 2021) was used to visualize the identified variants mapped on (1) a linear protein and its domains (“lollipop” plot); and (2) three-dimensional (3D) protein structures. Ensembl GRCh37 Release 104 (grch37.ensembl.org; accessed on 6 May 2021) was used to identify protein domains and the start and end positions of amino acids of LBD across transcript variants of *PPARG*. COSMIC (catalogue of somatic mutations in cancer) GRCh37 v94 database (cancer.sanger.ac.uk/cosmic; accessed on 6 May 2021) [24] was used to identify patients with thyroid carcinoma harboring CDS (coding DNA sequence) mutation in LBD of the *PPARG*.

### 2.4. MSKCC-ATC and TCGA-THCA Data

Nonsynonymous mutation data and clinical data were obtained directly from the cBioPortal (cbioportal.org; accessed on 28 May 2021) for poorly-differentiated and anaplastic thyroid cancers (MSKCC, JCI 2016) and thyroid carcinoma (TCGA, PanCancer Atlas). The R TCGAbiolinks package [25] (v2.18.0) was used to extract gene expression (RNA-seq) data from the cancer genome atlas thyroid cancer (TCGA-THCA). Read counts were normalized by trimmed mean of M-values (TMM) using the R edgeR package [26] (v3.32.1) and were subjected to the voom function in the limma package [27] (v3.46.0). The MutationAligner web resource (mutationaligner.org; accessed on 31 May 2021) was used to explore variants in “Hormone_recep” domain of *PPARG* in TCGA cohorts across 22 different tumor types [28]. The PathwayMapper tool (pathwaymapper.org; accessed on 31 May 2021) was used to export a curated cancer pathway image with alteration frequencies overlaid as scalable vector graphics from the cBioPortal (TCGA-THCA) [29].

### 2.5. MMD-THCA Data

Merged microarray-acquired Data (MMD) were generated for THCA comprising non-tumor (NT), ATC, PDTC, and PTC tissues, as previously carried out for other major cancer types [30,31,32,33]. The minimum information about a microarray experiment (MIAME) compliant datasets were carefully selected using gene expression omnibus (GEO), a public functional genomics data repository (ncbi.nlm.nih.gov/geo; accessed on 28 May 2021), based on the following criteria: (1) raw data availability (in CEL files); (2) tissue type annotation (i.e., NT, ATC, PDTC, and PTC); and (3) data derivation from affymetrix human genome U133 Plus 2.0 (HG-U133_Plus_2) array. Raw data of four selected GEO datasets (GSE29265, GSE33630, GSE65144, and GSE76039) were processed and normalized with robust multi-array average (RMA) using the R Affy package [34] (v1.68.0). These processed independent datasets were merged and corrected for batch effects using the ComBat function in the R sva package [35] (v.3.38.0). The R umap package (v0.2.7.0) was used to (1) visually identify technical (i.e., non-biological) variation derived from different studies and (2) validate the batch effect removal in ComBat-transformed MMD-THCA in low-dimensional uniform manifold approximation and projection (UMAP) space.

### 2.6. ScRNA-Seq Data

Expression matrices with molecule counts per gene per cell index of PTC tissues from 14 patients were obtained directly from GEO under the accession code GSE158291 [36]. For QC and preprocessing of scRNA-seq data, cells having unique feature counts over 10,000 or less than 200 counts and genes having less than 10 molecules across the cells were filtered out. The filtered data were used to perform normalization, feature selection (i.e., identification of highly variable genes), linear transformation (i.e., scaling the data), dimensional reduction (i.e., principal component analysis), cell clustering, and non-linear dimensional reduction (i.e., UMAP) using the R Seurat package [37] (v4.0.1). Data from nodular goiter were excluded from further analyses. Cell type identity was assigned manually to each cluster based on differentially expressed features using the FindAllMarkers function (Appendix A). Expression levels of the identified cell cluster-specific markers were assessed in a scRNA-seq study named “ICA: Ileum Lamina Propria Immunocyte (Sinai)” through the Single Cell Portal (singlecell.broadinstitute.org; accessed on 31 May 2021).

### 2.7. DE, GSEA, and GO Analysis

Differential expression (DE) analyses were performed using the linear modeling features of the R limma package [27] (v3.46.0). Statistical cutoffs of |log2FC| > 1 and adjusted *p*-value < 0.05 were applied to determine genes to be differentially expressed in both TCGA-THCA and MMD-THCA data. Gene set enrichment analysis (GSEA) was performed using the R fgsea package [38] (v1.16.0) to assess the enrichment of “GOBP_THYROID_HORMONE_GENERATION” gene set, which was downloaded from the Molecular Signatures Database (MSigDB), in PPARGi^high^ tumors. Gene ontology (GO) analysis and the enrichment analysis of disease-gene associations were performed using the enrichGO function and the enrichDGN function in the R clusterProfiler package [39] (v3.18.1), respectively.

### 2.8. CIBERSORT

The proportion of immune cell types in TCGA-THCA and MMD-THCA data was estimated using CIBERSORT (cibersort.stanford.edu; accessed on 1 July 2021). All PTC tissues were included in the analysis. The default LM22 (22 immune cell types) gene signature was used for each run with 100 permutations. The correlation between the estimated composition of immune cell types and PPARGi was assessed using the Pearson product-moment correlation test in the R stats package (v4.0.3).

### 2.9. ROC and Survival Analysis

The receiver operator characteristic (ROC) area under the curve (AUC) and the most optimal threshold (i.e., the threshold with the highest sum of sensitivity and specificity) were computed to evaluate the diagnostic accuracy of the PPARGi in deciding the presence of the disease using the R pROC package [40] (v1.17.0.1). Univariate Cox proportional hazards (PH) regression and Kaplan–Meier (KM) survival analyses were performed using the R survival package (v3.2.11). Samples with missing either disease-specific survival (DSS) data or gene expression data were excluded from the analysis. The regression coefficients, the Wald statistic *p*-values, hazard ratios (HR), confidence intervals (CIs) of the HR, and log-rank statistics of PPARγ target genes in predicting DSS in TCGA-THCA are listed in Appendix A. As previously described [30,32,41], the optimal cut-off used in the survival analysis was determined using Cutoff Finder [42] (molpathoheidelberg.shinyapps.io/CutoffFinder_v1; accessed on 31 May 2021).

### 2.10. PPARGi Derivation

The PPARgene database (ppargene.org; accessed on 28 May 2021), an open-source resource that curated the experimentally verified PPAR-α, -β/δ, and -γ target genes [43], was leveraged to query previously validated PPARγ target genes. Among the retrieved PPARγ target genes, only those annotated with “human” and “up” for “species” and “regulation”, respectively, were selected (Appendix A) for determining their prognostic role in predicting DSS in TCGA-THCA. Genes having the Wald statistic *p*-value of less than 0.05 were considered as prognostic and were used to construct a new prognostic index named PPARGi. PPARGi is computed by the sum of the expression level that is multiplied by predefined Cox PH regression coefficient of each PPARGi gene. Expression heatmaps of PPARGi-comprising genes were generated by Morpheus (software.broadinstitute.org/morpheus; accessed on 28 May 2021).

### 2.11. Machine Learning Algorithms for Disease Selection

Orange (v3.29.3) was used for t-SNE visualization, evaluation of ML models, and generation of confusion matrices and ROC curves. The schematic workflow is shown in Appendix A. The processed TCGA-THCA, MMD-THCA (PTC), and MMD-THCA (ATC) data were sent to the ‘test and score’ widget, in which multiple ML models were tested with the following defined parameters: (1) k-nearest neighbors (kNN, number of neighbors = 5, metric = Euclidean, weight = uniform); (2) support vector machine (SVM, cost = 1.00, kernel = RBF, numerical tolerance = 0.001, iteration limit = 100); (3) neural network (neurons in hidden layers = 100, activation = ReLu, solver = Adam with regularization = 0.0001, maximum number of iterations = 200); (4) logistic regression (regularization type = lasso (L1), strength = C1); and (5) random forest (number of trees = 10, subsets smaller than 5 were not split). These five models were evaluated using 10-fold cross-validation. The evaluation results are summarized in Table 1, Table 2 and Table 3 using the following defined parameters: (1) AUC = area under the ROC; (2) CA = classification accuracy, defined as the proportion of correctly classified examples; (3) F1 = weighted harmonic mean of precision and recall; (4) precision = proportion of TP among instances classified as positive; and (5) recall = proportion of TP among all positive instances in the given data. For cross-platform analyses, the ML models trained on TCGA-THCA data, which were TDM-transformed using the R TDM package (v0.3), as recently shown [33], were applied to MMD-THCA data consisting exclusively of PTC. 

## 3. Results

### 3.1. Targeted NGS of Thyroid Cancer-Related Genes in Monozygotic Twins with PTC

Formalin-fixed paraffin-embedded (FFPE) tissues derived from three pairs of identical twins with PTC, denoted as TP1, TP2, and TP3, were subjected to targeted next generation sequencing (NGS) for 103 thyroid cancer-related genes (see Section 2.2 of Materials and Methods; Appendix A). A total number of variants observed in twins varied from 98 (TP1) to 155 (TP2), in which the pairwise concordance varied from 34.5% (TP3) to 69.0% (TP2) (Appendix A). The SnpEff-annotated effects of the identified variants were mostly intronic variants, while other variants including 3′ and 5′ UTR variants, disruptive in-frame deletion, missense variants, synonymous variants, upstream and downstream gene variants, 5′ UTR premature start codon gain variants, and splice region variants were detected at lower frequency (Appendix A).

Notably, all three pairs of twins with PTC had four common variants, of which three (75%) occurred in the *PPARG* gene (Figure 1A) and the other variant occurred in the *TERT* gene. Using the MutationMapper at cBioPortal (see Section 2.3 of Materials and Methods), we found that two of the variants of *PPARG* (chr3: 12468710 and chr3: 12470239) occurred within the intronic region located between exon 5 and exon 6 of the *PPARG* gene encoding the “Hormone_recep” domain (PF00104, domain source: Pfam), which is defined as ligand-binding domain (LBD) of nuclear hormone receptor (Appendix A). These genomic positions mapped onto the previously reported 3D structure of the intact PPAR gamma–retinoid X receptor (RXR) alpha (PPARγ-RXRα) nuclear receptor complex (PDB identifier: 3E00) are shown in Appendix A. Thyroid carcinomas harboring the CDS mutations in the LBD of *PPARG* were further found in COSMIC GRCh37 v94 database (Appendix A) [24].

Of the SnpEff-annotated variants, those predicted with moderate impact in at least one of the twin pairs were found in *ARID1A*, *ALK*, *MSH2*, *FN1*, *KMT2A*, *TUBA3C*, *BRAF*, *STRN*, *KMT2D*, and *NF1* (Appendix A). The first twin pair (TP1) had the greatest number of missense variants in these genes, including an in-frame deletion in *MSH2*. They were discordant for missense variants in *TUBA3C* (*p*. Tyr262Cys) and *BRAF* (*p*. Val640Glu). For TP2, concordant variants were found in *BRAF* (*p*. Val640Glu) and *STRN* (*p*. Val620Leu), while missense variants in *KMT2D* (*p*. Gln827His) were present in only one of the twins. TP3 harbored missense variants in *FN1* (*p*. Asn172Asp) and *NF1* (*p*. Glu836Ala), which was present only in one of the twins who had recurrent cancer. To explore the frequency of nonsynonymous mutations in these identified genes in thyroid cancers, we leveraged cBioPortal-derived MSKCC and TCGA data (see Section 2.4 of Materials and Methods) for thyroid carcinoma (TCGA-THCA, *n* = 500) and poorly differentiated and anaplastic thyroid cancers (MSKCC-ATC, *n* = 117). Except for *BRAF* mutation occurring in 60% and 37% of PTC and ATC, respectively, nonsynonymous variants of *ARID1A, ALK, MSH2, FN1, KMT2A, TUBA3C, STRN, KMT2D*, and *NF1* were found in about 0.2–1.4% and 2.6–6% of patients in TCGA-THCA and MSKCC-ATC, respectively (Appendix A). These variants include putative driver mutations (in-frame variant, missense variant, and truncating variant) as well as amplification and structural variants (Appendix A). The PathwayMapper tool provided by cBioPortal [29] further identified RTK-RAS pathway as the most altered signaling pathway with alterations in *ALK*, *BRAF*, and *NF1* (score = 3.00; Appendix A).

### 3.2. Clinical Significance of Pparγ Target Genes in Thyroid Cancer

To examine the diagnostic and prognostic significance of target genes of PPARγ, in which all pairs of twins with PTC harbored the intronic variants, we next shortlisted the experimentally verified PPARγ target genes using PPARgene database (see Section 2.10 of Materials and Methods). Previous works have reported upregulation of the identified targets by PPARγ in human-derived tissues or cells (Appendix A). Univariate Cox regression survival analyses of TCGA-THCA-derived RNA-seq revealed that of the 39 targets, 10 PPARγ targets (25.6%) were favorable prognostic factor for disease-specific survival (DSS; Appendix A). To construct a personalized scoring system for patient classification, we developed an index termed as PPARGi, which is computed based on the regression coefficient and expression level of the 10 PPARγ target genes (Figure 1B,C). PPARGi varied greatly across tumor samples (*n* = 505) and was significantly higher than that of normal tissues (*n* = 59; Wilcoxon *p*-value = 2.2 × 10^−16^; Figure 1D). Using the area under the receiver operating characteristic (ROC) curve (AUC), we found that PPARGi distinguishes cancers from normal tissues with the AUC of 0.876 (Figure 1E), demonstrating the potential diagnostic use of PPARGi in deciding the presence of the disease.

PPARGi^high^ (*n* = 458; 93.5%) and PPARGi^low^ (*n* = 32; 6.5%) tumors stratified by the optimal cut-off index determined by Cutoff Finder (see Section 2.9 of Materials and Methods) had significantly different DSS outcomes (Figure 1F; log-rank *p*-value = 5.8 × 10^−16^). Through differential expression (DE) analysis between the two stratified groups, we found a total of 1775 DE genes with fold change (log2-base) > 1 and adjusted *p*-value < 0.05, as shown in Figure 1G. Intriguingly, gene set enrichment analysis (GSEA) revealed the enrichment of genes related to thyroid hormone generation (GO:0006590) in PPARGi^low^ tumors (normalized enrichment score = 1.72 and adjusted *p*-value = 0.008), which were found to highly express *DIO1, TPO, IYD, DIO2, TG, DUOXA2, FOXE1, PAX8, SLC5A5*, and *DUOX2* (Figure 1H). Additionally, gene ontology (GO) analysis of downregulated genes revealed “T cell activation” and ”regulation of T cell activation” as top-enriched terms in PPARGi^low^ tumors (Figure 1I). These findings are further corroborated by the CIBERSORT analysis, which identified regulatory T cells (Treg) as one of the immune cell types, in which their composition is positively corelated with PPARGi (Figure 1J). In addition to Treg, relative abundance of M0 and M2 macrophages, resting dendritic cells, and activated mast cells increased with PPARGi, while that of CD4 naïve T cells, monocytes, activated natural killer (NK) cells, eosinophils, plasma cells, and T follicular helper (Tfh) cells decreased (Pearson’s correlation *p*-value < 0.05).

### 3.3. Validation of PPARGi in MMD-THCA

To validate diverse expression levels of PPARγ target genes and their potential diagnostic value in thyroid cancer, we next generated a unified, cancer type-specific, merged microarray dataset (MMD-THCA), as previously carried out for other major cancer types [30,31,32,33]. Briefly, four independent transcriptomic datasets (GSE29265, GSE33630, GSE65144, and GSE76039) derived from affymetrix human genome U133 Plus 2.0 (HG-U133_Plus_2) array were normalized, integrated, and corrected for batch effects (see Section 2 Materials and Methods). These datasets comprised samples derived from anaplastic thyroid cancer (ATC; *n* = 52), poorly-differentiated thyroid cancer (PDTC; *n* = 17), papillary thyroid cancer (PTC; *n* = 69), and normal tissues (*n* = 78). As depicted in UMAP representation (Figure 2A), the newly developed MMD exhibited an overlay of samples colored by the source of data and a clear separation between samples derived from different tissue type (i.e., ATC, PDTC, PTC, and normal), demonstrating successful removal of batch effects arising from different studies. Technical validity of the ComBat-transformed MMD was further examined through DE analysis of PTC and normal tissues, in which the enrichment analysis of disease-gene associations (see Section 2.7 of Materials and Methods) identified “carcinoma, papillary”, “follicular adenoma”, “thyroid gland follicular adenoma”, and “follicular thyroid carcinoma” as top enriched terms in PTC (Figure 2B).

MMD-THCA exhibited heterogeneous expression of PPARGi-comprising genes, as previously seen in TCGA-THCA (Figure 2C). Consistent with our findings in TCGA-THCA, PPARGi of thyroid cancers was significantly higher than that of normal tissues (Wilcoxon *p*-value ≤  0.0001), of which PTC showed the highest mean of PPARGi among thyroid cancers (Figure 2D). Further, PPARGi achieved the AUC of 0.760, 0.654, and 0.899 in distinguishing cancers from normal tissues in ATC, PDTC, and PTC, respectively (Figure 2E). To validate the enrichment of genes related to thyroid hormone generation (GO:0006590) in PPARGi^low^ tumors observed in TCGA-THCA, we next stratified MMD-PTC into two groups, such that the proportion of PPARGi^low^ tumors in MMD (*n* = 5; 7.25%) would be comparable to TCGA stratification. DE analysis identified a total of 549 DE genes (Figure 2F), of which the query gene ontology term was enriched in PPARGi^low^ tumors (normalized enrichment score = 1.64 and adjusted *p*-value = 0.012), which were found to highly express *DIO1, TPO, IYD, PAX8, DIO2, FOXE1, DUOXA2, TG*, and *SLC5A5* (Figure 2G). While we did not find the correlation between PPARGi and the CIBERSORT-estimated proportion of Tregs previously observed in TCGA-THCA, resting dendritic cells and M2 macrophages repeatedly showed statistical significance (Pearson’s correlation *p*-value < 0.05), highlighting their robust association with PPARGi-comprising genes (Figure 2H).

### 3.4. Single-Cell Analysis of Ppargi-Comprising Genes in PTC

To identify specific cell types expressing 10 PPARGi-comprising genes in PTC, we next processed and analyzed a scRNA-seq dataset (GSE158291) derived from PTC tissues from 14 patients. Seurat (v4.0.1) identified a total of 15 cell clusters in PTC comprising 4045 QC-passed cells, including pericytes, plasma cells, group 1 innate lymphoid cells (ILC), T cells, B cells, endothelial cells, fibroblasts, epithelial cells, and macrophages (Figure 3A). Each cell cluster was defined manually using differentially expressed features (Figure 3B, Appendix A), in which the expression levels of immune cell type-specific features were validated using an independent scRNA-seq dataset obtained from the Single Cell Portal (see Section 2.6 of Materials and Methods). Notably, PPARGi-comprising genes were expressed predominantly in epithelial cells and macrophages (Figure 3C,D). While *GPD1*, *CYP27A1*, and *REN* were rarely expressed in PTC, the remaining genes including *DBI, APOE, SAT1, CDKN1A, KLF4,* and *PLIN2* showed highly heterogenous levels of expression contributing to varied range of PPARGi (Figure 3E). GO analysis of PPARGi-comprising genes identified lipid localization and transport, cholesterol/sterol/steroid catabolic process, and phospholipid transport as top enriched terms (Figure 3F), consistent with the known and putative roles of PPARγ in governing lipid storage, glucose, and insulin process [12].

### 3.5. Machine Learning for Disease Selection and Risk Stratification

To evaluate the diagnostic performance of expression profiles of PPARGi-comprising genes, we applied different machine learning (ML) algorithms to TCGA-THCA and MMD-THCA data annotated with the origin of tissue (see Section 2.11 of Materials and Methods; Appendix A). As PPARGi was derived from PTC expression data, cancers (TT) separated clearly from normal (NT) tissues in both TCGA-THCA (*n* = 564) and MMD-THCA (PTC, *n* = 147) and to a lesser extent in MMD-THCA (ATC, *n* = 130), as shown in Figure 4A. Of the tested ML models, LASSO multinomial logistic regression (LASSO) achieved the AUC of 0.990 in classifying TT from NT tissues in TCGA-THCA, demonstrating near-perfect diagnostic performance (Table 1 and Figure 4B). Similarly, thyroid tissues were best classified into TT and NT with SVM (AUC = 0.937) and LASSO (AUC = 0.945) in MMD-THCA PTC and ATC, respectively (Table 1 and Figure 4B).

We next aimed to demonstrate the clinical utility of PPARGi genes across different profiling platforms by developing the ML model from RNA-seq-acquired TCGA-PTC data and applying it to the microarray-acquired MMD-PTC data. Notably, a k-nearest neighbors (kNN) model developed from TDM-transformed TCGA-PTC data achieved the best AUC value of 0.942 in classifying TT from NT tissues in MMD-PTC, while the rest of the tested ML models achieved the AUC ranging from 0.890 (SVM) to 0.925 (neural network), showing the clinical applicability of PPARGi genes regardless of profiling platform (Table 2 and Figure 4C,D).

Lastly, we sought to evaluate the predictive potential of ML models in stratifying patients into PPARGi^high^ and PPARGi^low^ groups, which were found to have significantly different prognosis in terms of DSS (Table 3 and Figure 4D). The tested ML models achieved the AUC ranging from 0.965 (kNN) to 0.998 (SVM) in distinguishing the two risk groups, highlighting the clinical significance of PPARGi genes predictive of survival outcomes in thyroid cancer. These data altogether show the promising results of ML-optimized models built from expression profiles of PPARGi genes for clinical applications, which can easily be translated to other sequencing platforms.

## 4. Discussion

Despite the well-established role of the PAX8-PPARγ fusion oncogene, the expression landscape of experimentally validated PPARγ target genes remains unclear in thyroid carcinoma. In this study, we found that three pairs of monozygotic twins with PTC harbored four common variants, of which two variants occurred in the LBD of nuclear hormone in the *PPARG* gene, which have not been reported in healthy Korean population (Appendix A) [44]. We further showed that low expression levels of many of the PPARγ target genes were associated with poor survival outcomes, reinforcing the importance of PPARγ-RXRα pathway in governing immune microenvironment in PTC. Through scRNA-seq data analysis, we found that PPARGi-comprising genes were expressed most strongly in the epithelial cells and macrophages among the 15 cell clusters found in patient-derived PTC tissues. Lastly, ML models developed from RNA-seq-derived TCGA-THCA achieved a near-perfect performance in selecting the disease from benign tissues and in stratifying patients into the two risk groups in microarray-derived MMD-THCA, highlighting potential applicability of expression profiles of PPARGi-comprising genes in the clinical setting for patient management.

The LBD, situated in the C-terminus, is connected to the DNA-binding domain (DBD) via a flexible hinge region, which interacts physically with the DNA in PPARγ [11]. It is a key domain for transactivation and transrepression of PPARγ target genes that play important roles in adipogenesis, insulin sensitization, lipid metabolism, and inflammation, making PPARγ an effective target for the management of metabolic diseases, such as type 2 diabetes, obesity, and atherosclerosis [22,45]. The *PPARG* gene has 15 transcripts (splice variants), of which four transcripts lack exon 5 and 6, which encode LBD (Appendix A). Skipping of *PPARG* exon 5, for example, induced by ligand-mediated PPARγ activation, has been proposed as an alternative splicing event regulating PPARγ activity and PPARγ-related diseases [12].

In cancer, the tumorigenic role of PPARγ remains highly controversial. Studies have observed the inhibitory effects of PPARγ-RXRα signaling pathway on tumor growth, angiogenesis, differentiation, and that of production of inflammatory cytokines and tumor invasiveness, suggesting the anti-tumorigenic role of PPARγ in several cancer types including colon, lung, pancreas, prostate, and breast cancer [11,45]. By contrast, the pro-tumorigenic role of PPARγ has been reported in a variety of cancers, such as bladder tumor, renal pelvic tumors, hemangioma, lipoma, skin fibrosarcoma, mammary adenocarcinoma, and hepatic tumors, through distinct molecular mechanisms activated by PPARγ ligands regulating cancer cell proliferation, angiogenesis, and metastasis [46].

The cancer-related loss-of-function PPARγ mutations have been found predominantly throughout the LBD of the *PPARG* gene with varying degree of impaired ability in inducing transactivation of target genes [11]. Here, we observed that PPARγ target genes were expressed at significantly higher levels compared with benign tissues, suggesting that the PPARγ activation might increase the risk of developing thyroid cancer. Further, among PTC tissues, the low expression levels of PPARGi-comprising genes were closely associated with poor survival outcomes, indicating that the loss-of-function PPARγ variants might deregulate tumor microenvironment, including immune cell-infiltration, through macrophages in PPARGi^low^ tumors. In line with the findings supporting the role of PPARγ-RXRα pathway in modulating immune cell-infiltration, genomic alterations of PPARγ-RXRα complex (i.e., PPARγ^High^/RXRα^S427F/Y^) induced evasion of immunosurveillance and partial resistance to immunotherapies in muscle-invasive bladder cancer [47].

Through analysis of microarray- and RNA-seq-generated, patient-derived, tissue-level gene expression and scRNA-seq data, we assessed the expression profiles of diagnostic and prognostic PPARγ targets comprehensively in a cell type-specific manner and identified specific cell populations contributing the most to the computation of PPARGi in PTC. The last decade has seen the emergence of PPARγ as a key regulator of inflammatory and immune responses particularly in monocytes and macrophages, with anti-inflammatory effects in several disease models and clinical studies [48,49,50]. These studies have suggested that, in macrophages, PPARγ represses pro-inflammatory genes such as TNFα, IL-1B, IL-6, IL-12 MCP-1, and MMP-9 through unique ligand-dependent transcriptional mechanisms [11,21,48,51,52,53]. We thus carefully speculate that the poor prognosis observed in PPARGi^low^ PTC tumors might be attributed in part to the suppressed repression of the production of these pro-inflammatory genes in which the expression levels of many of the genes are potential predictors for metastasis or shorted survival time in thyroid carcinoma [54,55].

By curating multiple independent GEO datasets comprising a total of 216 patient-derived PTC and PTC-free tissues, we generated a merged PTC-specific MMD annotated with clinical features across different subtypes of thyroid carcinomas, including anaplastic thyroid cancer, poorly-differentiated thyroid cancer, and papillary thyroid cancer. These unified data processed using a uniform R pipeline in this paper source would allow parallel cross-platform analyses with TCGA, as previously shown in other major types of cancer [30,31,32,33]. In this study, we further demonstrated the cross-platform compatibility of the newly developed PPARGi, providing promising results for its clinical application and our informatics pipeline, which can be readily translated to other sequencing platforms.

## 5. Conclusions

It remains to be investigated whether the identified intronic variants found in monozygotic twins in the LBD of the *PPARG* gene would induce loss-of-function effects on PPARγ or impaired PPARγ-RXRα signaling pathway. Further, our sample size was small and might only represent a small subset of PTC cohort in Korea, although the expression levels and prognostic performance of PPARGi-comprising genes were extensively validated using public transcriptomic databases. We thus aim to assess the predictive power of PPARGi in a larger validation patient cohort or in a prospectively conducted study. Altogether, the functional PPARGi personalized scoring system may represent a powerful and effective genomic tool to improve patient management in PTC.

## Figures and Tables

**Figure 1 cancers-13-05110-f001:**
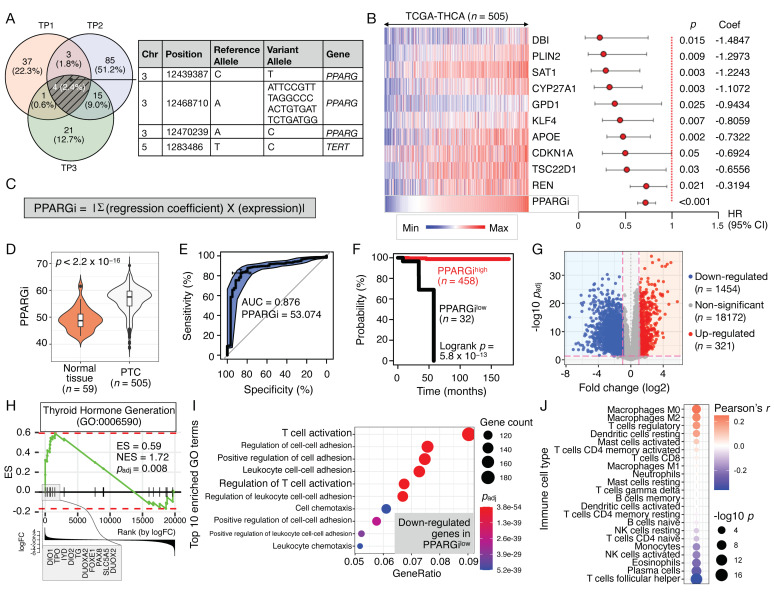
Diagnostic and prognostic performance of PPARGi in TCGA-THCA. (**A**) Venn diagrams showing the number of the SnpEff-predicted variants in three pairs of twins (TP1–3). Genomic positions of mutations, including chromosome (chr) number, and gene symbols are shown in the table (right). (**B**) Expression heatmap (left) and forest plot (right) of PPARGi-comprising genes. The data are sorted in increasing PPARGi. The horizontal axis of the forest plot represents hazard ratio (HR) with 95% confidence intervals (CI) estimated using a Cox proportional hazards model. The regression coefficients (Coef) and the Wald statistic *p*-values (*p*) are stated. (**C**) PPARGi computation for patient stratification. (**D**) PPARGi of normal (*n* = 59) and tumor tissues (*n* = 505). Mann–Whitney–Wilcoxon test *p*-values (*p*) and the number of samples (*n*) are stated. (**E**) The area under the ROC curve (AUC) of the PPARGi classifier. The AUC value and the optimal cut-off are stated. (**F**) The hazard ratio (HR), log-rank *p*-value, and the number of patients successfully stratified (*n*) determined from univariate Cox regression analysis are shown on the survival Kaplan–Meier (KM) curve. Black and red KM curves represent predicted PPARGi^low^ and PPARGi^high^ group, respectively. (**G**) Volcano plot depicting differentially expressed (red and blue) and non-significant (gray) genes in the PPARGi-stratified groups. The number of genes (*n*) are stated. (**H**) GSEA plot showing the enrichment of thyroid hormone generation gene set (GO:0006590) in PPARGi^low^ tumors. The cumulative enrichment score (ES) is plotted as the green curve, which is the running sum of the weighted ES as the analysis walks down the limma-generated ranked list. The vertical black lines on the horizontal axis of the plot indicate the position of query genes in the ranked list of genes. The bottom plot shows the value of the fold change (log2-base) as the computation goes down the limma-generated ranked list. Normalized ES (NES) and adjusted *p*-values (*p*_adj_) are stated. (**I**) Dot plot showing top gene sets (downregulated) in PPARGi^low^ tumors. (**J**) Dot plot showing Pearson’s correlations between PPARGi and CIBERSORT-estimated proportion of immune cell populations.

**Figure 2 cancers-13-05110-f002:**
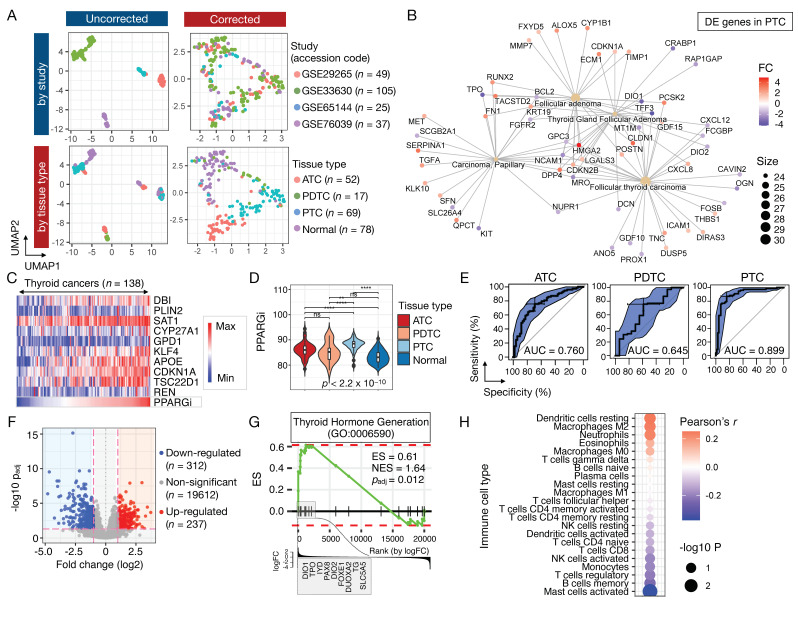
PPARGi in MMD-THCA. (**A**) UMAP representation of the uncorrected (left) and corrected (right) MMD. Data are colored by study (top) and tissue type (bottom). The number of samples (*n*) are stated. (**B**) Disease-gene associations depicting the linkages of genes and the enriched diseases as a network. The color and size of the node represent the value of fold change (FC) and gene count, respectively. (**C**) Expression heatmap of PPARGi-comprising genes. The data are sorted in increasing PPARGi. (**D**) PPARGi of normal and tumor tissues (ATC, PDTC, and PTC). The asterisks represent the statistical significance assessed by Mann–Whitney–Wilcoxon test (**** *p*  ≤  0.0001, ** *p*  ≤  0.01, ns: *p* > 0.05). Kruskal–Wallis *p*-value (*p*) is stated. (**E**) The area under the ROC curve (AUC) of the PPARGi classifier. The AUC values are stated for ATC (left), PDTC (middle), and PTC (right). (**F**) Volcano plot depicting differentially expressed (red and blue) and non-significant (gray) genes in the PPARGi-stratified groups in PTC. The number of genes (*n*) are stated. (**G**) GSEA plot showing the enrichment of thyroid hormone generation gene set (GO:0006590) in PPARGi^low^ tumors. The cumulative enrichment score is plotted as the green curve, which is the running sum of the weighted ES as the analysis walks down the limma-generated ranked list. The vertical black lines on the horizontal axis of the plot indicate the position of query genes in the ranked list of genes. The bottom plot shows the value of the fold change (log2-base) as the computation goes down the limma-generated ranked list. Normalized ES (NES) and adjusted *p*-values (*p*_adj_) are stated. (**H**) Dot plot showing Pearson’s correlations between PPARGi and CIBERSORT-estimated proportion of immune cell populations.

**Figure 3 cancers-13-05110-f003:**
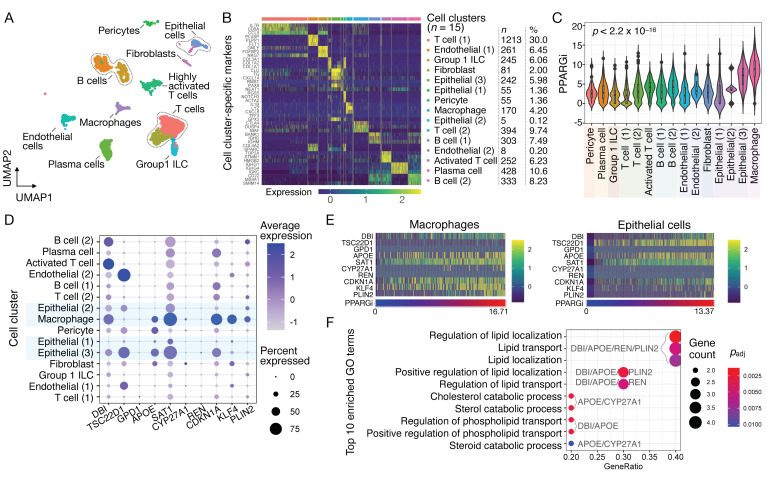
Analysis of PPARGi-comprising genes at the single-cell level in PTC. (**A**) UMAP of 4045 QC-passed cells depicting 15 cell clusters. (**B**) Heatmap of top 5 differentially expressed features across the identified cell clusters. The number (*n*) and proportion (%) of cells in each cluster are stated. (**C**) Distribution of PPARGi across different cell clusters. Kruskal–Wallis *p*-value (*p*) is stated. (**D**) Dot plot depicting average expression of PPARGi-comprising genes across different cell clusters. (**E**) Heatmap showing expression levels of PPARGi-comprising genes and PPARGi in macrophages (left) and epithelial cells (right). The data are sorted in increasing PPARGi. (**F**) Dot plot showing top enriched GO terms from PPARGi-comprising genes.

**Figure 4 cancers-13-05110-f004:**
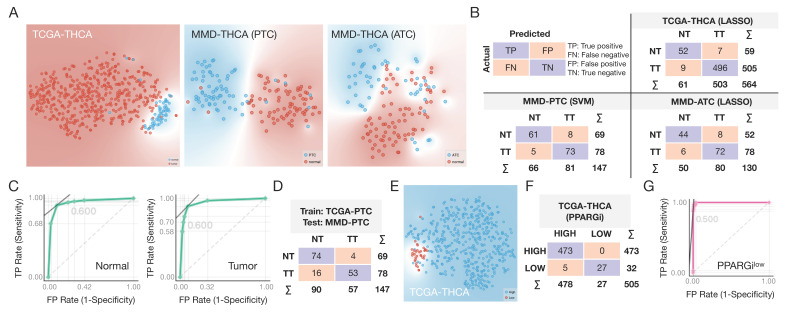
ML algorithms applied to expression profiles of PPARGi-comprising genes. (**A**) The t-distributed stochastic neighbor embedding (t-SNE) visualization of cancers (TT) and normal (TT) tissues in TCGA-THCA (left), MMD-THCA (PTC, middle), and MMD-THCA (ATC, right). (**B**) Confusion matrices of LASSO and SVM models in TCGA-THCA (top right), MMD-PTC (bottom left), and MMD-ATC (bottom right). (**C**) ROC curves and (**D**) confusion matrix showing classifying performance for normal (left) and tumor (right) tissues in MMD-PTC using kNN model developed from TDM-transformed TCGA-PTC. (**E**) t-SNE visualization of PPARGi^high^ (colored in blue) and PPARG^iow^ (colored in red) tumors in TCGA-THCA. (**F**) Confusion matrix of SVM model in classifying PPARGi^high^ from PPARGi^low^ tumors. (**G**) ROC curve showing classifying performance for PPARG^iow^ tumors.

**Table 1 cancers-13-05110-t001:** ML models for disease selection in thyroid cancer.

Dataset	Model	AUC	CA	F1	Precision	Recall
TCGA-THCA	kNN	0.980	0.961	0.962	0.963	0.961
	SVM	0.989	0.968	0.968	0.967	0.968
	Random forest	0.975	0.956	0.954	0.954	0.956
	Neural network	0.986	0.975	0.976	0.976	0.975
	Logistic regression	0.990	0.972	0.972	0.972	0.972
MMD-THCA (PTC)	kNN	0.925	0.884	0.884	0.885	0.884
	SVM	0.937	0.912	0.911	0.912	0.912
	Random forest	0.914	0.871	0.871	0.871	0.871
	Neural network	0.928	0.912	0.911	0.913	0.912
	Logistic regression	0.929	0.912	0.912	0.912	0.912
MMD-THCA (ATC)	kNN	0.944	0.838	0.835	0.841	0.838
	SVM	0.929	0.862	0.861	0.861	0.862
	Random forest	0.897	0.808	0.807	0.807	0.808
	Neural network	0.938	0.862	0.862	0.862	0.862
	Logistic regression	0.945	0.892	0.892	0.892	0.892

**Table 2 cancers-13-05110-t002:** Cross-platform evaluation results of the ML models.

Dataset	Model	AUC	CA	F1	Precision	Recall
Train-test data	kNN	0.942	0.864	0.862	0.873	0.864
	SVM	0.890	0.469	0.300	0.220	0.469
	Random forest	0.828	0.599	0.551	0.721	0.599
	Neural network	0.925	0.837	0.836	0.860	0.837
	Logistic regression	0.922	0.531	0.368	0.282	0.531

**Table 3 cancers-13-05110-t003:** ML models for risk stratification in thyroid cancer.

Dataset	Model	AUC	CA	F1	Precision	Recall
TCGA-THCA	kNN	0.965	0.980	0.979	0.979	0.980
	SVM	0.998	0.990	0.990	0.990	0.990
	Random forest	0.988	0.966	0.962	0.964	0.966
	Neural network	0.997	0.982	0.982	0.982	0.982
	Logistic regression	0.974	0.976	0.975	0.975	0.976

## Data Availability

Datasets used to generate MMD-THCA are available at GEO under the accession code GSE29265, GSE33630, GSE65144, and GSE76039. The scRNA-seq dataset analyzed in this study is available at GEO under the accession code GSE158291.

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
