# Peer review of "PPARγ Targets-Derived Diagnostic and Prognostic Index for Papillary Thyroid Cancer"

_cancers, 2021, doi:10.3390/cancers13205110_

Round 1
Reviewer 1 Report
In this paper, the authors study papillary thyroid cancer and develop a 10-gene personalized prognostic index based on gene expression of 10 PPAR_{\gamma}. Their machine learning algorithms show very good performance in deciding the presence of the disease and in selecting a small subset of patients with poor disease-specific survival in TCGA-THCA and a newly developed merged microarray data (MMD) comprising exclusively of thyroid cancers and normal tissues. The topic of the paper is interesting to the journal and the paper is well written. My concerns about this manuscript derive from the small sample size in the experimental validations, so I do not think I can recommend acceptance of this paper unless a major revision is provided, including a more convincing support of the results on larger and more diverse datasets.
Reviewer 2 Report
This study developed a PPARG targets-derived index, of which the authors demonstrated diagnostic and prognostic significance. In short, they found out that a higher PPARGi, which could be simply seen as up-regulation of PPARG target genes, is associated with a higher presence of macrophages, and better survival of papillary thyroid cancer patients. Machine learning models also confirmed the utility of the same set of target genes in disease selection and risk stratification. Below is a list of points for the authors.
- While you found PPARG mutations in the three pairs of twins, the mutation frequency in TCGA is not high. And no mutations were found in TCGA thyroid cancers. Is it because PPARG mutations are more likely to exist in certain populations? Did you know the mutation frequency in some other independent datasets of Korean patients? I am just curious. And while you provided good reasonings for studying PPARG in thyroid cancers in the Introduction, the sharp difference between your patients (6 actually) and TCGA somehow weakened the importance of studying PPARG. I’d suggest reducing the amount of description of Figure 1 and 2, and move some panels (for example, Fig 1A, 1B, and the whole Fig 2) into supplementary notes.
- Figure 1C. Are those mutations “intronic”? Then why does Figure 1D shows 3/4 are in protein domains? Is it because your annotations and the Lollipops used different versions of gene annotation files? You’d consider making them consistent.
- Figure 3E, Figure 4F. No points/dots in the figures.
- Figure 4B. No edges between the dots.
